# Role of Nutrition Information in Acceptance and Willingness to Pay for Biofortified Cereal Food: Implications for Better Health and Sustainable Diet

**DOI:** 10.3390/nu14163352

**Published:** 2022-08-16

**Authors:** Muhammad Rizwan, Azhar Abbas, Hui Xu, Umar Ijaz Ahmed, Ping Qing, Puming He, Muhammad Amjed Iqbal, Muhammad Aamir Shahzad

**Affiliations:** 1School of Economics and Management, Yangtze University, Jingzhou 434023, China; 2Institute of Agricultural and Resource Economics, University of Agriculture Faisalabad, Faisalabad 38040, Pakistan; 3Department of Agribusiness and Applied Economics, Muhammad Nawaz Shareef University of Agriculture Multan, Multan 66000, Pakistan; 4College of Economics and Management, Huazhong Agricultural University, Wuhan 430070, China

**Keywords:** awareness, biorfortification, bivariate analysis, food security, zinc deficiency

## Abstract

A range of nutritional needs are met through the use of fortified farm-based foods. Wheat biorfortification with zinc is such an example where biorfortification is carried out for a crucial element like Zinc. Zinc-biofortified wheat (Zn-wheat) has been officially launched in Pakistan since 2016 but its wide-scale dissemination, adoption and consumption have not taken place till to date. On the other hand, essential nutrients deficiencies have wide-ranging implications for public health especially for children and lactating mothers. This study is undertaken to know the reasons for the slow progression of scaling up of biofortified wheat varieties in Pakistan, people’s awareness about biofortified wheat and to recognize the role of information in acceptance and willingness to pay for this wheat. For this purpose, randomly selected 474 households were interviewed from four districts of Punjab province. They were categorized into four groups based on their exposure to information in real and hypothetical cheap talk (game theory context). Study findings reveal that respondents were ready to pay for fortified wheat if they are aware about nutrient aspects and Zn deficiency. Using Discrete Choice Experiment, the preferences for and factors affecting the willingness to pay for fortified wheat are evaluated. Main factors having positive impact include household head’s education and income, having pregnant women and children <5 years age. It was also found that people having valid information about nutrients of a food would be willing to pay more. The study highlights need for policy focus on educating people about nutritional aspects as well as making available biofortified foods to promote healthy living.

## 1. Introduction

Micronutrients refer to essential minerals and vitamins, obtained from the diet and fundamentally required to support all metabolic functions [1]. Micronutrient deficiencies (MNDs) can cause malnutrition and have an enormous negative impact on health, eventually leading to death if not overcome timely. A recent estimate shows that 2 billion people are affected by MNDs across the world, most of them belonging to developing countries mainly caused by low quality diet. Another cause of MNDs is driven by inadequate diets, the direct cause of inadequate diet intakes [2,3,4,5].

Many human ailments originate from poor quality and vitamin-deficient diet. For instance, the most micronutrient sicknesses originate from Zinc (Zn), vitamin A (Vit. A), iron (Fe), and iodine (I) deficiencies. Nevertheless, zinc deficiency has become more problematic in recent year [6,7,8,9,10] due to its intensifying impacts among human beings, especially in children, childbearing- and breastfeeding women, elderly fellows as well as plants [11,12,13]. According to a report published by the World Health Organization (WHO), Zn deficiency is considered as one of the five topmost challenges related to micronutrients in developing countries and eleventh in the world. Similarly, Vitamin A and Iron [14] have negative impact on an estimated one-third of the population worldwide [13,15].

Zn is a fundamental micronutrient for all living organisms [16,17]. It requires more than three hundred enzymes as a cofactor [18] and has an intensive structural role in various proteins, comprising many transcription factors [19,20]. Zn deficiency consequences were documented, for the first time, in the early 1960s for human health [21,22]. It significance for human body can be judged from the fact that it is the second plentiful component in the body, following iron [23]. As the human body does not have a storage system for Zn, its daily and optimal intake is necessary albeit the required intake is higher for growing children and pregnant women [24,25]. The main benefit of this micronutrient, however, is to improve the immune system against different diseases, particularly cold fever and pneumonia [26,27,28].

Albeit Pakistan has developed on many fronts, yet, an enormous proportion of population of the country is living under the poverty line and facing micronutrient deficiencies. At the same time, Zn deficiency has been elucidated as a crucial issue amongst micronutrient deficiencies [29]. As per the estimates of the Pakistan National Nutrition Survey, Zn deficiency is quite high in the country being most prevalent among children and women. In this regard, around 21 women of reproductive age (WRA) and more than 18 percent children are found to have Zn deficiency in the country [30]. To facilitate a global comparison, Figure 1 indicates zinc-deficient countries with different colors showing priority level, where Pakistan is ranked 23rd with top-level Zn deficiency among 128 countries (Figure 1). However, as noted by Gupta et al. (2020) [31], not all 128 countries have carried out plasm Zn concentration survey rather the map is based on stunting as a proxy indicator implying risk of zinc deficiency. 

UNICEF noted in its annual report-2016 that Pakistan has witnessed a higher children mortality rate globally as 89 children of every 1000 born alive and die due to malnutrition before reaching five years of age. Among these, 55 children with 61.8 percent of the total mortality ratio could hardly enter the second month of life [32]. Since last twenty years, children mortality rate has gone up by 8 percent in the country. The mortality rate can be controlled if sufficient nutrition is provided to the children as most of the deaths are caused by diarrhea [32].

Consistent with Pakistan’s Scaling-up Nutrition (PSUN) report, Pakistan has to spend a cost of US$ 7.6 billion being 3 percent of its gross domestic production (GDP) every year in a bid to fight malnutrition. This cost contains healthcare costs, laborers’ loss, and reduced productivity [33]. At the province level, Punjab province, an irrigated territory, always remained hub of green revolution for wheat crop where modern varieties are grown. Despite that fact, huge percentage of rural population faces malnutrition issues including Zn deficiency [29]. The consumption of zinc biofortified wheat can help eradicate Zn deficiency among low-income households who cannot buy enriched food products or Zn supplements [29]. This is much practicable as yearly per capita total wheat consumption is around 100 kg, providing daily per capita calorie intake of 920 kcal (equivalent to 37% of daily calorie intake per person) [34]. The data pertaining to energy intake with respect to various risk groups such as WRA, children and elderly could not be found for the country, On the other hand, according to USDA data, Zn-wheat is as needed due to wheat cultivation in low zinc soils in many countries including Pakistan [32]. Since the launch of Zinc wheat in Pakistan, its share remains only 0.1% of total wheat production [35]. Figure 2 shows the priority of zinc biofortified wheat programs in different countries, and Pakistan is ranked as 1st country in terms of the potential scope and positive impact of zinc biofortification [29].

The main aim of biorfortification is to improve human nutrition and health without decreasing the yield or damaging the health of the crop [36]. Cereals, belonging to a distinctive group of monocots family, are cultivated in many countries for edible food of the world’s population. Therefore, Zn-enriched grains would potentially enhance health status and reduce health risks [12]. Selective plant breeding, use of zinc fertilizers through foliar application, soil amendments for enhanced micronutrients’ uptake by plants, soil microbiome improvement, seed priming with micronutrients and genetic engineering are the major means to achieve biorfortification. A large-scale investment has been going on in many developing countries for its promotion however, the investment quantum is significantly less so far [37]. 

A low quantum of investment is perceived to be driven by lack of information on people’s acceptance for such foods. However, willingness to pay (WTP) has a significant role in eliciting the potential for mass uptake especially for economic valuation. WTP is a technique that is applicable for estimation of people/groups’ aptitude to understand the monetary value for specific medical treatment or intervention [3]. Thus, knowing people’s acceptance and their willingness to pay for any treatment like biofortified wheat would pave the way for effective policy formulation related to Zn-wheat.

For the promotion of mass production and consumption of biofortified wheat, there is a need to know its acceptance level among masses along with the level of their willingness to pay for such products once they enter the market [38,39]. A study conducted on Zn-biofortified wheat flour’s efficacy among women, children, and non-lactating women of childbearing age in India have revealed promising outcomes by the introduction of low zinc wheat flour (LZWF) and high zinc wheat flour (HZWF) to more than 6 thousand respondents [40]. Such a study is lacking in case of Pakistan (although, a significant work done by Aga Khan University in terms of survey and research is underway and is expected to be reported soon) where zinc deficiency is found on a large scale among the population, especially in children under five years and; childbearing and breastfeeding women. Furthermore, the poor (mostly women and girl children) do not have access to zinc-rich foods and zinc biofortified wheat. On the one hand, due to COVID-19, value chains of fruits, vegetables, animal-source food (i.e., perishable food) have disturbed, the poor are likely to have less access/affordability to nutrient-rich foods (e.g., animal source food). On the other hand, households are tempted to increase their consumption of non-perishable, staple foods. Hence, zinc biofortified wheat can play a significant role in ensuring nutritional security of people. In this regard, zinc biofortified wheat was introduced in Pakistan officially in 2016 but its widespread production and consumption generally remained subdued [41]. Little to no work has been done to explore such dynamics related biofortified food in the country except for some of the unpublished work by Aga Khan University, Karachi, who have completed a survey on, inter alia, WTP for and consume biofortified wheat. Hence, this study is conducted, firstly, to understand whether consumers have enough information or knowledge about zinc wheat and nutrition values linked with biofortified wheat. Secondly, the study tries to understand whether prior information about zinc deficiency and zinc-wheat awareness have an impact on consumers’ WTP. In addition, the study also aims to estimate the exact amount consumers are willing to pay more for reducing Zn deficiency through zinc-wheat along with the influencing factors for this WTP.

## 2. Conceptual Framework of the Study

Zinc-wheat is a novel food item to most consumers in Pakistan as data are not available in the market or any published work that has analyzed its acceptance, hence some method(s) need to be generated. Generally, systematized elicitation using revealed preference data through different conditional assessment techniques restricts respondents from observing real budget constraint as they may not have actual feelings of such preference/demand and would not behave the same. According to [42], a factor with three dimensions can overstate consumers’ preferences in a hypothetical setting. The data collected under this study avoid this problem by confirming that consumers have actual need for a product (here Zn-wheat) and face budget constraints accordingly. However, this study uses a hypothetical technique for comparison. Discrete choice experiments (DCEs) captures more curiosity by eliciting preferences stated through different ways. For example, Ref. [43] demonstrated the theory of consumer choice and revealed DCEs theoretically by stating that characteristics or features define commodities as treatments rather than things. Moreover, DCEs are based on behavioral random utility models (RUM) and hence are econometrically manageable [44].

DCEs are supportive techniques to quantify the eminence and discretion of several characteristics epitomized in a new food item. For instance, a higher price can be indicative of higher Zn content. Additionally, if a DCE takes up actual products and real money, such as in the present study, general issues of assumed preference can be overcome. However, participants can be biased if they do not know about new food and hence they would overstate existing one. In the present study, wheat is used as a staple food and is largely known to participants to aptly suit the application of DCE. Under this experiment, the participants considered selecting “j” as a substitute i.e., Zn wheat. Utility, which can be derivative by ith participant from choosing “j” alternative, comprise of two elements (i.e., random element and methodical element) and can be specified as:(1)$$U_{ij}=V_{ij}+\varepsilon_{ij}$$
where, $V_{ij}$ represents a share of utility’s function, a contingent characteristic of the product as well as a stochastic component. Supposing that participants maximize their utility by choosing “j” alternative because they can compare the products. So, a rational customer selects a substitute that maximizes his utility. By considering $Y_{i}$ as a random variable indicating outcomes of this selection/choice, the probability of individual $i^{\prime }s$ selection of j is denoted as under:(2)$$P\left(Y_{i}=j\right)=P\left(V_{ij}+\varepsilon_{ij}\right)>P\left(V_{ik}+\varepsilon_{ik}\right)\;\forall k=1,2,\ldots ,j;k\neq j$$

Taking insights from the existing literature, education/awareness about can be considered as a factor to influence consumers’ acceptance of biofortified crops [45,46,47]. In this regard, Lanou et al., illustrated in their study that education about nutrition among communities can play a vital role in enhancing nutritional diversity among children and women in Kenya [47]. Likewise, Okello et al., show that biofortified crops’ awareness improves the acceptance among nutrient-deficient populations [46]. Therefore, the present study hypothesized that awareness has a positive effect on willingness to pay (H1) for a nutrition-enhancing food such as biofortified Zn-wheat. Similarly socioeconomic factors could either influence such outcomes positively or negatively (H2) whereas the level of perceived utility positively impacts willingness to Pay (H3) (see Figure 3).

## 3. Research Methodology

### 3.1. Selection of Study Districts and Respondents

A survey was done from October 2017 to January 2018 in four districts: Multan, Khanewal, Rahimyar Khan, and Muzaffargarh of Punjab in Pakistan. Figure 4 shows the targeted area of the present study for obtaining the data to meet objectives of the study. The selected districts are in the South of the province, and the southern region is generally less-developed. The majority of the population is found undernourished particularly children under five years age and childbearing and pregnant women [48,49].

Four approaches corresponded to this study’s data collection drive while the number or respondents was almost same from each selected district and under each category of approach. Among the four approaches, the first one used “real with no information” to decide for purchasing wheat varieties, i.e., zinc biofortified and conventional one without any information about their nutrition values. With this approach, 122 respondents were targeted for this action. Under this approach, respondents were briefed about the availability of biofortified wheat in the market and were prompted to share their awareness about this product. This group was treated as a control group. The second approach adopted was the “real with information”, meaning that information about Zinc wheat’s nutritional values was given to 118 respondents. This approach is applied to evaluate the demand-pull strategy on willingness to pay because the amount of information may have a varied impact on decision-making through information provision [50]. 

Further, as applied in the contingent valuation studies, the hypothetical valuations may vary, so the third approach, "hypothetical without cheap talk (choice experiments generally employ such nomenclature as a means of communication between players/participants that do not directly affect the payoffs of the game while information provision and reception do not involve any cost thus having no impact of ultimate choice. Cheap-talk addresses the question of how much information can be credibly transmitted through direct and costless communication) was undertaken for 120 respondents by providing information regarding nutrition to them. The information provided to the respondents included the uses of and access options for biofortified wheat although they were not given a loaf of biofortified wheat. With the fourth approach, "hypothetical with cheap talk", 114 respondents were interviewed by informing them about the nutrition values. The survey sample size is described in Table 1.

### 3.2. Respondents’ Distribution vis-à-vis Approaches and Location

Table 2 illustrates data sampling of respondents about numbers and regions. Almost similar number of respondents were taken for the experiment in four villages of two districts, namely Khanewal and Muzaffargarh; while, two urban areas in Multan and Rahimyar Khan Districts. Respondents were selected randomly, and households head or spouses of household heads were interviewed in this experiment. Almost equal number of respondents were interviewed for each type of term/treatment both in rural and urban areas. Moreover, the participants’ interviews were completed in the primary market area. Among the total number of respondents, 67 percent were male-headed households while 33 percent female households head or spouses took part in the experiment. Wheat flour is used almost three times a day in meals as *Chapati* (Pakistani flat bread) within the whole country. Therefore, there is no conflict to buy wheat flour among households. Sometimes, respondents revealed they would change their meal decision and eat rice instead of *Chapati*. That decision was taken by male head among 52 percent household and by spouses of household head among 37 percent households’ while the decision to switch food/meal on the advice of the children of head was reported by 11 percent households.

### 3.3. Experimental Procedure 

To achieve the study objectives for each term used in the previous section, every respondent was asked to eat *Chapati* (a loaf of bread) made of flour of local wheat and zinc-wheat varieties. Zinc-wheat (Zincol-2016) flour was arranged from the local suppliers in/around the village. Afterwards, four attributes were used and noted as: (i) sensory acceptability, (ii) information provided to the respondent(s) for better understanding, (iii) choice experiment, and (iv) demographic variables method. Further, respondents were divided into four groups and selected randomly for groups 1 to 4, as shown in Table 1. Each respondent was given 100 PKR (equal to around 60 US. Cents as per exchange rate at the time of data collection) as an inclusion fee in the experiment. This approach is generally followed in many WTP studies [50]. 

#### 3.3.1. Sensory Acceptability

This procedure is adopted from [51] where each person had to taste at least one-fourth Chapati of each variety. Both types of Chapati were cooked in the same way as cooked traditionally. After eating both types of Chapati, they were asked to score each type using Likert scale coding with 1 = Like very much and 5 = Dislike very much in terms of taste and overall acceptance.

#### 3.3.2. Information about Nutrition

In this experiment, respondents were given nutrition information and asked whether they had any information about zinc-wheat before or not. For this purpose, three types of respondents were used for real, without, and cheap talk hypothetically. This step is used as a control variable in the analysis of the data.

#### 3.3.3. Choice Experiment

Under this experiment, face-to-face instructions were given to all respondents [52,53,54]. The adjustment was made in the instructions as required due to a change in commodity and mechanism of provision. A choice sheet was provided to each respondent who completed the experimental process with their choice. In the term “hypothetical”, respondents were told that it doesn’t mean that you must have to purchase it if you make a choice. Furthermore, it was articulated to respondents for the term used “real information” after completing all choice experiments where one situation can be selected as a choice to buy in real. The extra price for Chapati baked by zinc wheat flour was elicited from each respondent they were willing to pay in comparison with traditional wheat flour Chapati. Each respondent’s choice and the stated price were noted. Respondents had to make choices on the basis of taste and information given to them. Afterwards, the collected information under this experiment was designed following Lusk & Schroeder [55]. This technique is useful in countering multicollinearity problem as stated prices of products are not linearly related.

#### 3.3.4. Demographic Variables

Data regarding demographic variables (such as education, income, age, family size, etc.) were recorded during the experiment to analyze their impact on willingness to pay for zinc-wheat. Moreover, all participants were informed about the purpose of collection of this information before starting the experiment. It was explicitly made clear that experiment’s results will only be used for education and research purposes, and to do so, verbal consent was taken from each respondent to evade their hesitation to participate in between the interview. However, they were free to take part or not in the experiment. Only volunteers were recruited for the experiment after presenting the complete experimental design and procedure in the experts’ committee meeting, where it was confirmed and approved by the committee members.

### 3.4. Econometric Model

A bivariate probit model was applied to estimate the determinants of choice between tow varieties. The possible relationship among the choice decisions are as follows:(3)$$Y_{ij}=x_{ij}^{\prime }\beta_{j}+\varepsilon_{ij}$$
where *Y_ij_* (*j* = 1, …, m) indicates the Chapati varieties choices (thus, m = 2) faced by ith farmer (i = 1, …, n), the vectors that affect the adoption decisions for choice are given by $x_{ij}^{\prime }$, which is a 1 × k vector, the unknown parameter *β_j_* denotes a k × 1 vector to assess, and $e_{ij}$ indicates an unobserved error term. According to this description, each $Y_{ij}$ is a dichotomous variable, and, therefore, Equation (1) is a part of the estimated m equations (in this case, m = 2).
(4)$$Y_{1}^{*}=\alpha_{1}+X\beta_{1}+\varepsilon_{1}$$
(5)$$Y_{2}^{*}=\alpha_{2}+X\beta_{2}+\varepsilon_{2}$$

Hence, $Y_{1}^{*}$ and $Y_{2}^{*}$ are two dependent variables representing each acceptance decision of the Chapati choice such that *Y_j_* = 1 if $Y_{j}^{*}\;$> 0; 0 otherwise.

## 4. Results and Discussion

### 4.1. Demographic Characteristics

The summary statistics of respondents’ demographic characteristics is presented in Table 3. This table shows findings on four treatments used and information regarding zinc wheat and other characteristics of the participants such as income, family size, etc. Results show that there is not a significant difference amongst the four treatments used for individual preferences. It shows appropriateness of selection of four treatments. Our study results are supported by a previous research to analyze Africans’ willingness to pay [3]. Further, the four targeted regions are also included and illustrated in Table 3. Similarly, results on sensory factor i.e., taste also show that participants did not show a significant taste difference between both varieties, i.e., biofortified and conventional wheat.

### 4.2. Variety Specification and Price Effect

Table 4 shows the specifications of wheat varieties used in the experiment along with price effect. Results illustrate that there was a difference among participants regarding the selection of conventional and Zinc wheat varieties. However, there is not a significant difference in case of full sample. Responses to four treatments indicate that participants have many different choices and select more conventional rather than zinc wheat without information. But, when given information concerning Zinc impact on life, they change their choice. The results of previous study conducted in Kenya support the present results [46]. Further, the findings related to other two treatments used (i.e., hypothetical with and without cheap talk) also show that with cheap talk, the choice is opposite of that for without cheap talk (Table 4). The price effect negatively impacts, as demonstrated by log-likelihood about four treatments used in the model.

### 4.3. Willingness to Pay (WTP) and Marginal Willingness to Pay (MWTP)

Table 5 reveals the results of willingness to pay and marginal willingness to pay for zinc wheat in terms of four treatments applied in the experiment. A bivariate model was used to examine the effects, and results revealed that there is variation between WTP for two varieties of wheat. For conventional wheat, people do not want to pay more. It may be due to the fixed price announced by the government for wheat crop in the province, however, there is variation in each treatment for zinc wheat. Specifically, the WTP for zinc wheat is higher than conventional one for each treatment. This finding is supported by a study conducted in Kenya regarding willingness to pay for fortified cereal products [56]. Regarding treatments, the results are highly intuitive and show that participants were willing to pay more for zinc wheat without giving information but only 5 percent higher than the existing market price for conventional wheat. For the second treatment, i.e., real with information, WTP was higher than for the first treatment, i.e., 16 percent higher price compared with conventional wheat. Similar findings are reported by [57,58] who found that health information has a significant impact effect on the demand for better quality food among US citizens. Furthermore, these findings also correspond to another study analyzing WTP by [3]. 

Participants’ response on WTP for zinc wheat show that they are highly interested in this interventions and are ready to pay up to 27 percent higher price vis-à-vis conventional wheat under the treatment ‘hypothetical without cheap talk’. In case of Pakistan, a very recent study also stated that people are willing to accept biofortified Zn wheat flour in KPK province [59]. The last method of experiment with the treatment ‘giving information with cheap talk’ shows even more WTP and thus higher interest for Zn-wheat among households in comparison with other treatments/scenarios and the conventional wheat. A higher WTP due to prior or more detailed information about the intervention is well supported by the existing literature and thus underscores the significance of the role it plays in the mass uptake of a food item, intervention or idea. For instance, as per [60], male and female customers were shown ready to pay a premium, respectively, up to 26 percent and 49.3 percent for genetically modified food in Europe. In the similar vein, having awareness regarding vitamin A given to customers in Uganda was found to have significantly positive impact on WTP fortified maize [3].

Results imply that a higher WTP for zinc wheat correlates with more demand for bio-fortified zinc wheat. It reveals that farmers would be better placed if they focus on increasing production of zinc wheat. The present study results also show that it is difficult to increase the premium among low-income populations in developing countries. But, in the case of the present study, people are ready to pay more for zinc wheat possibly due to the face–to-face information given via about zinc, Zn-wheat and its nutritional values to the participants can potentially drive up this amount as evinced by [3].

### 4.4. Determining Factors of Willingness to Pay

A bivariate model was applied to examine the determining factors affecting participants’ WTP for zinc wheat, the results of which are given in Table 6. The results indicate that increase in the price of conventional wheat negatively affects consumers’ behavior as their WTP for it significantly decreases. However, the increase in conventional wheat price has a positive and significant impact on WTP for Zn-wheat. It may be due to the provision of information about the importance of Zinc to the participants. Similarly, education positively influences WTP for zinc wheat but it has non-significant impact on WTP for conventional wheat. This finding is intuitive as well, because most people are aware of the conventional wheat being major source of staple food and any improvement in formal/informal education would not significantly affect people attitude towards higher payment for the existing food. Nevertheless, with higher education, willingness to pay increased for zinc wheat. It shows the awareness among participants after getting nutritional information. Earlier research also illustrated that education positively impacts wiliness to pay for food [37,46,56,61].

Additionally, results show that family size has a significant negative impact on conventional wheat’s WTP while there is a negative but non-significant impact Zn-wheat’s WTP. Increase in family size would lead towards higher conventional wheat demand and people would find it hard to meet increased expenditure on Zn-wheat. However, smaller families can afford to pay more for nutrition-rich food as it would be required in relatively smaller quantity for such families. It further implies that an increase in the number of family members affects families’ purchasing power for traditional variety. Although the large family size negatively affects WTP for biofortified wheat, households would still want to buy but cannot buy due to their budget constraint. Studies by [3,37,62] widely report such outcomes in their results.

Furthermore, results show that participants/households having children under five years age have a significant impact on WTP for Zn-biofortified wheat after getting nutrition information. However, there is no significant impact on conventional wheat variety, although it has a positive value. It shows being the staple food, respondents must have to buy wheat. Furthermore, variables on breastfeeding and pregnant women are also positively significant for zinc wheat regarding willingness to pay. It reveals that pregnant and breastfeeding women in the households showed their interest to pay more for their better health after they received nutrition information. This finding is further supported by the findings of [63] who report that zinc uptake among women has increased by a range of 3-6 mg in Pakistan by using Zincol-16. The results also indicate that income has a positive impact on zinc wheat implying that increased income would improve purchasing power and hence their WTP for nutritional food like biofortified wheat along with their urge to improve family and personal health. Moreover, biofortified wheat preference over conventional one regarding taste is not significant, which shows that respondents do not find major taste difference between two varieties. Overall, in all districts (rural and urban), the impact is positive for WTP for biofortified wheat varieties. It indicates that information has a substantial influence on consumers’ willingness to pay for biofortified wheat. A previous study showed that different socio-economic factors influence the acceptance of new food [64]. Variable on gender of the household head is shown to have a non-significant impact on the WTP for both wheat varieties. This is also intuitive and justified as wheat has to be consumed by whole family, and not individually, hence anyone deciding the consumption of staple food has to keep in mind whole family’s preferences.

## 5. Conclusions and Policy Recommendations

Zinc deficiency persists among communities and individuals in many developing countries, particularly among children, pregnant, and childbearing women. The income constraints and lack of resources are main causes of food and nutritional insecurity among lower-income households. In the present study, willingness to pay for zinc-biofortified wheat was examined and compared with conventional wheat varieties in four districts of Punjab province, Pakistan. As the zinc biofortified wheat is recently launched officially in Pakistan, there is not much data available to analyze WTP for its use, hence in the present study, an experimental process generated the data. Participants were interviewed face-to-face before and after given information regarding nutrition, especially about biofortified wheat, which reduces zinc deficiency among people. Still, this crop’s success mainly depends on its success in the market based on net profitability for farmers and consumers’ acceptance and how much they would pay for such food. Therefore, knowing implications of biofortified wheat prices on a macro level could be a significant step to understand the consumer’ behavior and make decisions about the crop. In the present study, a unique model was used to analyze the impact of different factors on the willingness to pay for zinc wheat. For this purpose, real money was given to the respondents to take part in the experiment.

This study shows that participants are willing to pay more for zinc biofortified wheat than the conventional one. It also proved that consumers, with no given information about Zinc and nutrition, are less willing to pay than the other three treatments used, i.e., real giving information, hypothetical with cheap talk, and without cheap talk including information provided to the respondents. Moreover, the study analyzed the impact of information given to participants. Results showed that participants were ready to pay up to 27 percent more for Zn-wheat after they are given information about Zn deficiency and ways to handle it. The bivariate model results indicate that many factors were positively significant for willingness to pay about zinc biofortified wheat (such as children under five years, pregnant and breastfeeding women, education, income etc.) within the household. Hence, education has a positive impact, and more educated consumers are willing to pay more. Sensory factors are essential to accept a new food. The taste factor was analyzed in this study, and the result did not show an impact regarding taste. It shows that consumers do not find any taste differences between the two varieties. Similar implications hold for zinc-biofortified wheat flour as many people are still unaware of the health benefits and the nutritional aspects of micronutrient-fortified cereal flour in the country. For this, promotion of such products by highlighting their health and nutritional benefits would greatly improve their uptake in the short run and food security in the long run.

Given the study results, the following main key policies are being recommended, which need to be focused. Firstly, as awareness about zinc deficiency mediates more demand and entices households to pay more for biofortified wheat, people need to be made aware of the uses and availability of biofortified wheat and similar products to increase their nutrional security. A higher WTP for biofortified wheat implies, at the same time, a promising market for such products. Here, one thing to be noted is that the price of wheat in Pakistan is supportive to farmers and, on the other hand, controlled for customers. Hence, this mechanism of price may not affect the poor’s purchasing power.

Secondly, more educated, having more income, having children under the age of 5 years, and pregnant women households are willing to pay more; therefore, on the one hand, institutions and stakeholders need to enhance education and resources for income generation in the targeted population. On the other hand, policies should focus to promote zinc wheat among poor people who need to improve their health and immune system as they are already deficient in health and income. For this, targeted subsidy can be provided as they cannot afford the higher prices for novel and healthy food, even if it is a staple food. Thirdly, as the large family size has a negative and significance impact on WTP for Zn-wheat, this shows the dwindling purchasing power when family grows in number. For this, campaigns need to be initiated, and if underway already, need to be geared up to control population. These options would help enhance the micronutrient and improve the immune system among poor people in developing countries to defeat numerous diseases. The policy also need to promote further research that should analyze the effective ways of awareness about Zn deficiency, which is a limitation of our study. 

## Figures and Tables

**Figure 1 nutrients-14-03352-f001:**
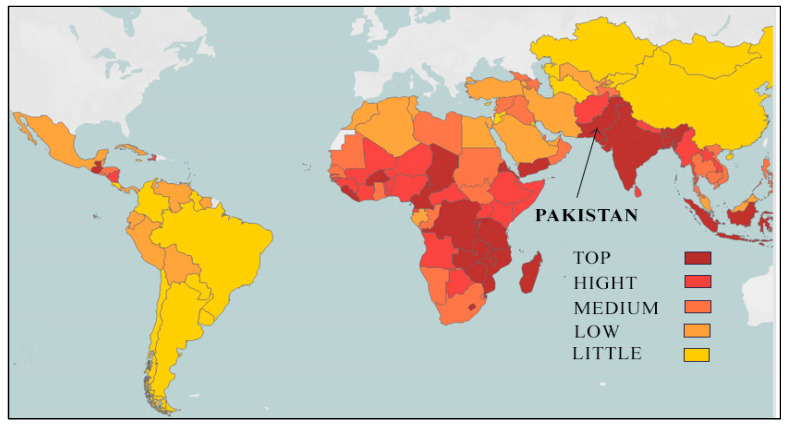
Countries with zinc deficiency (source harvest plus): available at https://bpi.harvestplus.org/subindex_micronutrients.html?id=c2 (accessed on 5 April 2022).

**Figure 2 nutrients-14-03352-f002:**
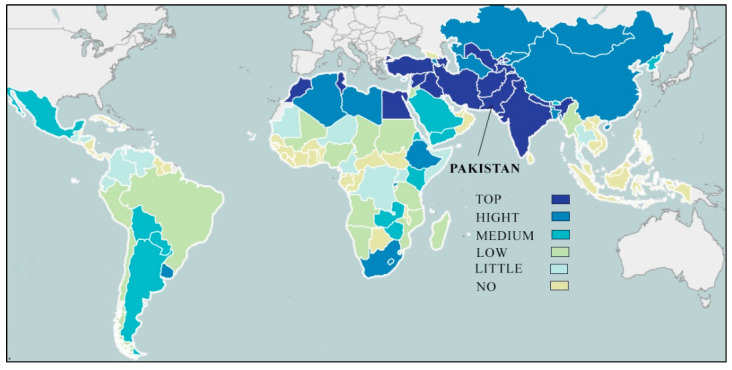
Zinc-wheat priority level (source harvest plus): available at https://bpi.harvestplus.org/bpi_cropmaps.html?id=c8 (accessed on 28 March 2022).

**Figure 3 nutrients-14-03352-f003:**
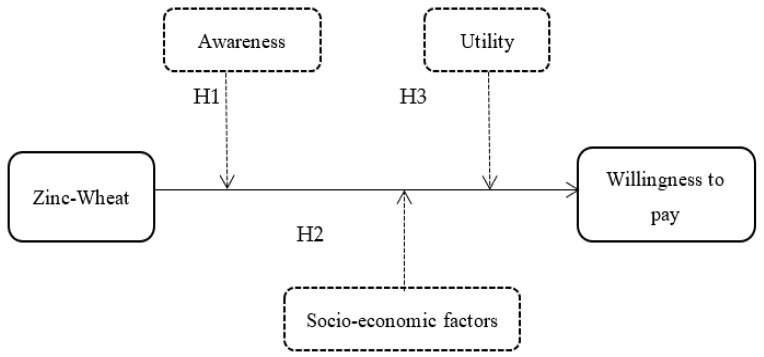
Conceptual model employed by the study.

**Figure 4 nutrients-14-03352-f004:**
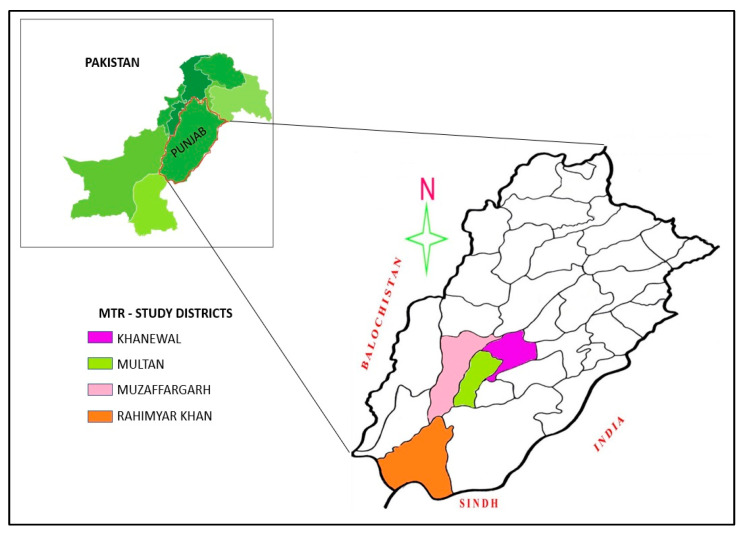
Map of Punjab province showing study districts.

**Table 1 nutrients-14-03352-t001:** Survey Design.

	Real	Without Cheap Talk—Hypothetical	With Cheap Talk—Hypothetical
No information regarding Nutrition	1	--	--
Given information about Nutrition	2	3	4

**Table 2 nutrients-14-03352-t002:** Distribution of respondents with respect to approaches and location.

Area/Region	Districts	Not Given Information	With Given Information			Total
		Real	Real	Hypoth. without Cheap Talk	Hypoth. Cheap Talk	
Rural	Khanewal	31	29	30	28	118
	Muzaffargarh	29	32	30	29	120
Urban	Multan	32	27	32	30	121
	Rahimyar Khan	30	30	28	27	115
Total		122	118	120	114	474

**Table 3 nutrients-14-03352-t003:** Summary Statistics of socioeconomic variables under each treatment.

Used Variables	Descriptions of Variables	Full Sample	Not Given Information	With Given Information		
			Real	Real	Hypoth. without Cheap Talk	Hypoth. with Cheap Talk
*Taste*	Respondent’ preference between varieties					
Conventional wheat	% of respondents who chose conventional wheat	48.2	51.6	50.7	49.6	50.2
biofortified wheat	% of respondents who chose zinc wheat	52.8	49.4	49.3	51.4	49.8
*Demographic and income variables*						
Gender	%age of male	0.57 (0.005)	0.43 (0.0213)	0.498 (0.021)	0.449 (0.012)	0.471 (0.017)
Education	Schooling years	7.287 (0.051)	6.241 (0.0437)	7.957 (0.0489)	7.124 (0.053)	6.987 (0.047)
Family size	Number of family members	6.213 (0.059)	5.789 (0.079)	6.137 (0.081)	6.241 (0.021)	6.021 (0.071)
Children <5 yrs	Number of children under 5 years	1.318 (0.021)	1.298 (0.039)	1.495 (0.026)	1.369 (0.093)	1.387 (0.024)
Breastfeed/pregnant	Number of breast-feeding/pregnant women	0.372 (0.006)	0.395 (0.019)	0.323 (0.013)	0.309 (0.015)	0.401 (0.016)
Income	Household income per year-PKR	279,000 (253,121)	253,612 (24,846)	312,420 (268,913)	251,024 (264,555)	302,180 (243,544)
Prev. inform	%age of respondents who have information before experiment	0.224 (0.003)	0.201 (0.009)	0.198 (0.010)	0.291 (0.012)	0.299 (0.12)
*Location*						
KWL	Khanewal district	118	31	29	30	28
MNT	Multan district	120	29	32	30	29
RYK	Rahim Yar Khan district	121	32	27	32	30
DGK	Dera Ghazi Khan district	115	30	30	28	27

**Table 4 nutrients-14-03352-t004:** Estimation of parameter—bivariate probit model.

	Full Sample	Not Given Information	With Given Information		
		Real	Real	Hypoth. without Cheap Talk	Hypoth. with Cheap Talk
Specific constant of varieties					
Conventional wheat	5.2134 (0.3245)	9.2341 (1.3024)	3.8476 (0.4972)	3.7210 (0.9870)	8.3649 (1.0254)
Zinc wheat	4.3627 (0.3102)	4.7261 (0.8617)	5.1278 (0.6321)	5.9742 (0.9421)	6.2171 (0.9941)
Price effect regarding own					
Conventional wheat	−0.0425 (0.0023)	−0.3641 (0.0621)	−0.0571 (0.0021)	−0.0142 (0.0079)	−0.0312 (0.0082)
Zinc wheat	0.0391 (0.0041)	−0.0092 (0.0004)	−0.0510 (0.0047)	−0.0049 (0.0006)	−0.0094 (0.0009)
Log-likelihood	−5431.43	−892.61	−925.96	−1412.02	−903.39

**Table 5 nutrients-14-03352-t005:** Willingness to pay (WTP) and Marginal Willingness to pay (MWTP) for conventional and Zinc wheat based on the bivariate probit model.

	Not Given Information	With Given Information		
	Real	Real	Hypoth. without Cheap Talk	Hypoth. with Cheap Talk
Total willingness to pay				
Conventional wheat	90 (7.4015)	90 (6.5324)	90 (8.2513)	90 (7.2341)
Zinc wheat	95 (9.2359)	105 (11.2508)	115 (11.4186)	108 (10.5268)
Marginal willingness to pay				
Zinc wheat vs. conventional	5 (5%)	15 (16%)	25 (27%)	18 (20%)

**Table 6 nutrients-14-03352-t006:** A bivariate probit estimation of correlates of willingness to pay – information with real.

	Variety	
	Conventional Wheat	Biofortified Wheat
Price of conventional wheat	−0.00421 ** (0.0012)	0.00024 (0.0004)
Price of zinc wheat	0.00024 (0.0035)	0.00067 * (0.0002)
Gender	0.00521 (0.4561)	0.00211 * (0.1120)
Education	0.08721 (0.0371)	0.02371 ** (0.0312)
Family size	−0.04102 ** (0.0420)	−0.31207 (0.0517)
Children <5 yrs	0.23866 (0.2356)	0.34502 ** (0.0689)
Breast feed/ pregnant	0.04213 (0.0412)	0.76852 * (0. 3514)
Income	0.00524 (0.1023)	4.96584 * (0.0239)
Taste-preference	0.51225 (0.0681)	0.84534 (0.1354)
Prev. inform	−1.38916 (0.3816)	−0.82347 (0.0612)
KWL	−0.94263 (0.0681)	0.57630 * (0.0325)
MNT	−0.74233 (0.0281)	0.47031 * (0.1320)
RYK	−0.64063 (0.1681)	0.07630 ** (0.2115)
DGK	−0.84001 (0.2981)	0.43330 ** (0.0624)
Constant	3.94528 * (0.2205)	5.23404 *** (0.0952)
Log-likelihood	−869.21350	

Note: *, **, *** respectively stand for significant at 1%, 5% and 10% level of significance.

## Data Availability

Data will be shared to the interested individuals/organization upon request by the first author.

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
