# Peer review of "Role of Nutrition Information in Acceptance and Willingness to Pay for Biofortified Cereal Food: Implications for Better Health and Sustainable Diet"

_nutrients, 2022, doi:10.3390/nu14163352_

Round 1

Reviewer 1 Report

I am not able to read through the paper as the English is just not of sufficient quality. I got stuck in the methodology, and I could not figure out the design of the study. Please have the paper thoroughly checked by an English copy-editor and resubmit.

Author Response

We are highly thankful to the learned reviewer for his/her time to read the paper and feel sorry for any inconvenience due to the weakness in presentation. We have thoroughly revised the paper in all aspects starting form abstract to conclusion. We have completely streamlined the flow, presentation and structure of the paper for easy and lucid reading by the learned reviewer.  The paper has been completely revised both technically (by the authors) and in terms of English (by English Expert being a native English speaker).

Reviewer 2 Report

The authors provided the data by taking survey from four districts from Pakistan Punjab province in relation to “Role of Nutrition Information in Acceptance and Willingness to Pay for Biofortified Cereal Food: Implications for Better Health and Sustainable Diet,” Various factors affected the acceptability of the biofortified wheat flour and positive changes were brought into the knowledge of the respondents once information was given on the aspect of biofortified wheat flour. The study conducted by the authors is important in the view that Zn is highly deficient in soil and food produced on it and making aware to the consumers is essential once the economic product of the crop come in to the market for consumption. The authors have written the MS very well giving the valid justifications and adopting proper statistical methods for drawing the valid conclusions. However, there are some changes which are required to be done before the final decision on the submitted MS. Authors can see comments in the MS (pdf comments) and give answer to queries. Please see the corrections and comments in the PDF MS (give answer to the queries). Regarding various section like introduction, methods and materials, Results and discussion are written well. Some more relevant references are required to be put in the MS as shown in the MS.

Reviewer 3 Report

Dear authors, 

I enjoyed reading your submitted article. It is of high significance, and kudos to the authors for doing a job well. There are significant grammar/spelling errors, and the illustration resolution is very low. Makes for a tedious reading.

Line 21: 'Place' misspelt.

Line 24: Mediating is not the correct word usage.

Line 33-34: Restructure

Line 34: Replace 'education' with 'educating'

Line 40: Micro functions?

Line 46/154/219/258/321/367/370: Extra spacing between words

Line 48: Use 'originate' instead of 'start'

Line 62: Separate it and is

Line 64: Replace 'higher' rather than 'increases'

Line 68: replace 'albeit' with 'although'

Fig.1. Poor resolution. Reconstruct the image rather than copying from Harvest Plus

Line 82: No need to expand UNICEF. Commonly accepted abbreviation. Also does not seem like it was included in the paper elsewhere?

Lines 82-85: Restructure

Line 96: Replace 'practice' with 'consumption'

Line 104: Insert ref.

Fig 2: Poor resolution. Reconstruct

Line 116: Change 'significance' to 'significant'

Line 119: Insert reference where it says reference

Line 121: Missing period at the end of the sentence

Lines 122-124: Restructure

Lines 136-137: Restructure, confusing

Lines 139-142: Split into two sentences

Line 147: Predilection is a fairly heavy word usage. Try preference.

Line 221: Restructure

Please provide what kind of hypothetical info was provided

Line 262: Explain why respondents in the 'real with no info' did three treatments while others did all four.

Line 264: Explain chapati cooking procedure

Line 274/283: Add space between words

Line 283: No need for year of the reference

Lines 363-365: Restructure

Line 368: 'In' and not 'Ln'

Line 382: One quotation mark at the end. Remove

Line 393: Willingness is misspelt

Line 431: Main causes of what?

Line 433: So the Zn-fortified wheat flour is recently launched but consumption has not started yet as noted in the abstract, correct?

Line 445: Willing instead of willingness

Lines 460-463: Split into two sentences, restructure

Lines 479:  Covid-19, irrelevant, remove
